# Comparative Characterization of Immune Response in Sheep with Caseous Lymphadenitis through Analysis of the Whole Blood Transcriptome

**DOI:** 10.3390/ani13132144

**Published:** 2023-06-29

**Authors:** Jitka Kyselová, Ladislav Tichý, Zuzana Sztankóová, Jiřina Marková, Kateřina Kavanová, Monika Beinhauerová, Michala Mušková

**Affiliations:** 1Department of Genetics and Breeding of Farm Animals, Institute of Animal Science, 104 00 Prague, Czech Republic; tichy.ladislav@vuzv.cz (L.T.); sztankoova.zuzana@vuzv.cz (Z.S.);; 2Department of Genetics and Breeding, Faculty of Agrobiology, Food and Natural Resources, Czech University of Life Sciences Prague, 165 00 Prague, Czech Republic; 3Department of Microbiology and Antimicrobial Resistance, Veterinary Research Institute, 621 00 Brno, Czech Republic; jirina.markova@vri.cz (J.M.);

**Keywords:** *Corynebacterium pseudotuberculosis*, gene expression, immunogenetics, RNA-seq

## Abstract

**Simple Summary:**

Caseous lymphadenitis is a contagious bacterial disease that affects both domestic and wild animals. It has been studied by scientists since the 19th century. Thanks to new high-throughput RNA sequencing techniques, we can now better understand how the immune system responds to this disease, evaluate individual resistance levels, and identify disease biomarkers. This knowledge can help prevent the spread of incurable diseases, reduce economic losses, and breed farm animals with greater bacterial resistance. In this study, we explore the potential role of immune genes in the fight against disease in ewes from an ordinary farm. The ovine genome reflected host–pathogen interactions by promoting transcriptional changes in innate and acquired immunity in diseased sheep. Furthermore, in exposed sheep, innate immunity increased over the adaptive response.

**Abstract:**

Caseous lymphadenitis (CL) is a chronic contagious disease that affects small ruminants and is characterized by the formation of pyogranulomas in lymph nodes and other organs. However, the pathogenesis of this disease and the response of the host genome to infection are not yet fully understood. This study aimed to investigate the whole blood transcriptome and evaluate differential gene expression during the later stages of CL in naturally infected ewes. The study included diseased, serologically positive (EP), exposed, serologically negative (EN) ewes from the same infected flock and healthy ewes (CN) from a different flock. RNA sequencing was performed using the Illumina NextSeq system, and differential gene expression was estimated using DESeq2 and Edge R approaches. The analysis identified 191 annotated differentially expressed genes (DEGs) in the EP group (102 upregulated and 89 downregulated) and 256 DEGs in the EN group (106 upregulated and 150 downregulated) compared to the CN group. Numerous immunoregulatory interactions between lymphoid and nonlymphoid cells were influenced in both EP and EN ewes. Immune DEGs were preferentially assigned to antigen presentation through the MHC complex, T lymphocyte-mediated immunity, and extracellular matrix interactions. Furthermore, the EP group showed altered regulation of cytokine and chemokine signaling and activation and recombination of B-cell receptors. Conversely, NF-kappa B signaling, apoptosis, and stress response were the main processes influenced in the EN group. In addition, statistically significant enrichment of the essential immune pathways of binding and uptake of ligands by scavenger receptors in EP and p53 signaling in the EN group was found. In conclusion, this study provides new insights into the disease course and host–pathogen interaction in naturally CL-infected sheep by investigating the blood transcriptome.

## 1. Introduction

Caseous lymphadenitis (CL) is a contagious chronic bacterial disease of global relevance that affects small ruminants, horses, cattle, pigs, wildlife, and even humans. It is also known as pseudotuberculosis or “cheesy gland.” The disease is caused by a Gram-positive facultative intracellular parasite called *Corynebacterium pseudotuberculosis* (*Cp*). It belongs to the vitally important CMNR group encompassing *Corynebacterium*, *Mycobacterium*, *Nocardia*, and *Rhodococcus* [1,2,3]. It is often referred to as “iceberg disease” because it can cause a gradual decline in physical condition, thin ewe syndrome, fertility problems, behavioral changes, decreased milk and wool production, and poor carcass quality [4,5,6,7]. Unfortunately, the disease is challenging to eradicate due to difficulties in detecting subclinically affected animals, poor response to antibiotic treatment, and the ability of *C. pseudotuberculosis* to persist in different environments while evading the host’s immune system [8,9].

Sheep infected with *Cp* experience a slow progression of encapsulated pyogranulomas, which eventually lead to the formation of larger abscesses in superficial or visceral lymph nodes and organs, such as the lungs, liver, and kidneys [10,11]. The main factors contributing to the virulence of *Cp* are its cell wall lipids and exotoxin phospholipase D, which promote bacterial persistence and spread within the host and result in macrophage death and chronic inflammation [12,13,14]. Researchers are currently conducting genomic and proteomic studies to identify *Cp*’s antigenic potential and pathogenicity islands involved in evading the host’s immune system. These studies aim to produce more effective vaccines for preventing infection and limiting emerging antibiotic resistance to reduce CL outbreaks on farms [5,15,16,17]. However, the cost of commercial vaccines remains high, and they have yet to provide full protection against infection or effective control of CL.

The reasons why some sheep and goats are more susceptible to *Cp* than others, whether due to physiological or genetic factors, are still not fully understood. However, studies suggest that a cellular immune response, specifically a Th1 response, plays a significant role in controlling the infection, as the microorganism is facultatively intracellular. The formation of pyogranulomas, which are characteristic of *C. pseudotuberculosis* infection, involves both humoral and cell-mediated immunity, a complex process that requires the activation of macrophages and T lymphocytes [18,19,20]. Recent advances in high-throughput genomic technologies have enabled the first comprehensive profiling of the goat spleen transcriptome and proteome in response to experimental infection with *Cp* [21,22,23]. Currently, genome-wise approaches and the generation of big data are considered promising selection-associated techniques that can provide animal farming industries with the ability to cope with the issues caused by diseases through breeding for health traits [24].

According to our findings (unpublished), cases of CL have increased in the Czech Republic over the last decade, affecting an average of 20% of animals in approximately 30 tested sheep and goat flocks. To better understand the disease and host defense, we studied global gene expression in sheep infected with *Cp*. The study’s main goals were the comprehensive characterization of the ovine whole blood transcriptome in the later course of the infection in naturally infected ewes. Moreover, we performed a functional comparison of the differentially expressed genes between healthy and affected animals based on their differing serological statuses. Thus, this research provides new insights into the host immune response against *Cp* in sheep.

## 2. Materials and Methods

### 2.1. Animals and Sampling Procedure

The study included nine five-year-old females of the East Frisian breed from two different farms located in Czechia. The animals had been maintained under comparable housing conditions and nutritional regiments and were regularly orally dewormed with First Drench (Virbac, Milperra, Australia) at the recommended dose rate of 1 mL/5 kg body weight (BW) or Aldifal of 1 mL/10 kg BW dose (Mikrochem, Pezinok, Slovak Republic). A commercial ELISA serological test (Elitest CLA, HYPHEN Biomed, Neuville-sur-Oise, France) was carried out to detect any IgG antibodies against phospholipase D, which could indicate the presence of disease. This direct enzyme-linked immunosorbent assay (ELISA) utilizes a recombinant form of the important, conserved *C. pseudotuberculosis* virulence factor, phospholipase D (PLD) to detect anti-PLD IgG antibodies in sera from sheep and goats with CL. As PLD is not known to be produced by any other sheep pathogenic bacteria this makes it a very specific test. When sera from infected animals are screened with the ELISA, antibodies specific for PLD bind to the recombinant antigen and are quantified. Elitest CLA is the most commonly used test because of its cost efficacy and acceptable test performance (sensitivity 87%, specificity 98%), [7]. The confirmation test for *Cp* presence was performed by classic bacterial cultivation. The first experimental group consisted of the three serologically negative ewes exposed to *Cp* infection, which did not show CL clinical signs (group exposed—negative; EN). The second experimental group from the same flock consisted of the three diseased, positive ewes (group diseased—positive; EP) with visible abscesses. Finally, the third control group included three healthy females from the other noninfected farm (group control—healthy; CN). Peripheral blood samples (3 mL) were drawn from the jugular vein into Tempus™ Blood RNA Tubes (Thermo Fisher Scientific, Waltham, MA, USA), immediately cooled and then stored at −80 °C until RNA isolation.

### 2.2. RNA Isolation, Library Preparation, and Sequencing of the Transcriptome

Total RNA was extracted from the cryopreserved stabilized whole blood of ewes using a Tempus™ Spin RNA Isolation Kit (Thermo Fisher Scientific) according to the manufacturer’s instructions. RNA quantity and quality were assessed by a NanoDrop 1000 spectrophotometer and Qubit 4 fluorometer (Thermo Fisher Scientific). The RNA integrity was verified using an Agilent Bioanalyzer 2100 (Agilent Technologies, Santa Clara, CA, USA). All samples fulfilled the criteria of a 260/280 ratio between 2.0 and 2.2 and RNA integrity numbers greater than 7; the average yield was 114.7 ng/µL of RNA. The mRNA-seq libraries were constructed per the manufacturer’s instructions using the NEBNext^®^Ultra™ II Directional RNA Library Preparation Kit (New England Biolabs, Ipswich, MA, USA). The libraries were pooled equimolarly and then sequenced once on an Illumina NextSeq 550 system lane following the vendor’s protocol using Mid Output Kit v2.5 and PhiX Sequencing Control Kit v3 (Illumina Inc., San Diego, CA, USA). We achieved a total of 257,619,546 paired-end reads, with a sequencing read range of 30,875,434 to 35,323,270 bp. On average, 32 million 150 bp paired-end reads were obtained per sample.

### 2.3. Transcriptome Data Processing, Differential Expression Analysis, and Gene Ontology Classification

Raw sequencing data were automatically processed by the Basespace cloud interface (Illumina Inc., USA) in default settings. The basecalling, adapter clipping, and quality filtering were carried out using bcl2fastq v2.20.0.422 Conversion Software (Illumina Inc., USA).

The raw reads in FASTQ format were cleaned from low-quality reads and adapter sequences using Trim Galore (v0.5.0) software (https://github.com/FelixKrueger/TrimGalore). The software was used in default mode, which includes trimming low-quality ends from reads (with a default Phred score of 20) and removing the Illumina universal adapter sequence AGATCGGAAGAGC. We set the maximum allowed error rate at 0.1 and discarded any reads that were shorter than 20 bp after quality or adapter trimming.

Further analysis was divided into two parts: novel transcript prediction and differential expression analysis. Transcriptome reference (GCF_002742125.1_Oar_rambouillet_v1.0_rna.fna) was downloaded from the NCBI database. First, the program bowtie2 (v2.3.4.3) (https://github.com/BenLangmead/bowtie2) in default mode was used to map samples against the reference and predict new transcripts. Any unmapped reads were extracted using SAMtools (v0.1.19-44428cd) (https://github.com/samtools/), and assembled by program SPAdes (v3.1.0.) (https://github.com/ablab/spades). The new predicted contigs were then cleaned for possible contamination using blast (v2.5.0+) (https://ftp.ncbi.nlm.nih.gov/blast/executables/blast+/v2.5.0+/). The cleaned contigs were compared to *Ovis aries* transcriptome and genome reference and to the human genome, viral, bacterial, and rRNA database (SILVA db) using blast (v2.5.0+). Finally, Trinotate (v3.1.1) (https://github.com/Trinotate/Trinotate/releases)was used to annotate the cleaned contigs. In the second part of the analysis, the trimmed samples were aligned to the transcriptome reference, which included both the downloaded reference and the newly predicted contigs from the first part of the analysis. Mapping and quality statistics were performed using QualiMap (v.2.2.2-dev) (https://github.com/EagleGenomics-cookbooks/QualiMap; Eagle Genomics, Cambridge, UK) in the Multisample BAM QC mode [25]. The count of uniquely mapped reads was extracted and saved into a matrix using the featureCounts (v 1.6.5) program (https://github.com/topics/featurecounts) [26]. The count matrix was created based on transcripts.

Transcriptomic data were divided into three groups of sheep (EN, EP, and CN) based on their serological status. We used two different approaches, DESeq2 (v 1.26.0) (https://github.com/topics/deseq2 ) and Edge R (v 3.28.0) (https://github.com/StoreyLab/edge), to perform differential expression analysis. DESeq2 corrects for library size and RNA composition by using a method called median of ratios, while Edge R uses a normalization method called trimmed mean of M-values (TMM). We compared the transcript expression between all animal groups (CN × EN, CN × EP, and EN × EP) and used the Benjamini–Hochberg (BH) method for *p*-value adjustment in DESeq2 or the false discovery rate (FDR) in Edge R. Transcripts with a log2 fold change greater than 1.5 or less than −1.5 and a *p*_adj_ < 0.05 or FDR < 0.05 were considered significantly differentially expressed (DE).

The sets of DEGs were interpreted using the Gene Ontology (GO) system with the online software tool WEB-based GEne SeT AnaLysis Toolkit, WebGestalt 2019, which allowed for classification of enriched cellular components, biological processes, and pathways implemented in the KEGG (Kyoto Encyclopedia Genes and Genomes), Reactome, and Wiki Pathway databases [27]. Due to a noticeable lack of functional data for *Ovis aries*, human or cattle genomes need to be used as references. The BH correction for a *p*_adj_ (FDR) value of 0.05 and the top 10 significance levels were set as the thresholds for significantly enriched GO terms and pathways. Redundancy reduction was realized as the weighted set cover approach, which finds top gene sets while maximizing gene coverage.

### 2.4. Reverse-Transcriptase Real-Time qPCR Validation

To verify the accuracy of the high-throughput sequencing data, four DEGs with higher opposite expression, *MFA5* (microfibril-associated protein 5), *MEDAG* (mesenteric estrogen-dependent adipogenesis), *HMCN1* (hemicentin 1), and *LY6G6C* (lymphocyte antigen 6 family member G6C), were chosen for analysis via real-time quantitative PCR to confirm their transcriptional levels. Reverse transcription was carried out using the High-Capacity cDNA Reverse Transcription Kit according to the manufacturer’s instructions (Thermo Fisher Scientific). The relative mRNA levels were analyzed using 2-step RT qPCR with specific predesigned TaqMan Gene Ovine Expression Assays (Thermo Fisher Scientific) consisting of primers and fluorescently labeled probes (Appendix A). The short TaqMan probes were designed to span exon–exon junctions to avoid contamination of genomic DNA. As an optimal reference, *B2M* (beta 2 microglobulin), *POLR2A* (RNA polymerase II subunit A), and *PPIB* (peptidylprolyl isomerase B) genes were selected (geNorm average stability value M = 0.25). qPCR measurements were performed in triplicate using a LightCycler 480 II Instrument (Roche Diagnostics, Basel, Switzerland, EU). The reaction mixture was prepared in an 8 µL volume and consisted of the TaqMan Gene Expression Assay, 480 LightCycler Probes Master Mix (Roche Diagnostic, Mannheim, Germany, EU), and 2 µL of 5× diluted cDNA. Experimental data analysis was performed using qBase+ (v3.2) Premium software (Biogazelle, Ghent, Belgium). The relative gene expression was estimated using the generalized qBase model of delta-delta-Ct, which takes into account gene-specific PCR efficiency correction and multiple reference gene normalization [28].

## 3. Results

### 3.1. Sequencing Quality Overview

We used high-throughput RNA deep sequencing to analyze the global expression profiles of sheep with different CL serological statuses in the affected flock compared to the control, healthy animals. Eight RNA samples were successfully sequenced, and one sample of the EN group failed sequencing quality control. The basic sequencing statistics of the reads are shown in Table 1. After removing adapters and sequences of low quality, an average of 31.94 mil reads per library was finally obtained. Alignment to the ovine reference yielded mean values per library of 26.5 million reads in positive ewes, 24.2 mil reads in negative ewes, and 19.9 mil reads in uninfected controls, representing 63.64 to 78.29% of the cleaned reads. The rate of uniquely mapped reads was lower, reaching a maximum mean of 57% in the control group, but the sample coverage indicated that every base of reference was, on average, represented more than 21 times in the reads. The reads mapping to multiple regions were excluded from downstream DE analysis. The GC content was determined to be between 41% and 51%, meeting the necessary analytical criteria. We obtained 27,430 transcripts and 12,230 genes, covering 54.8% and 43.3% of known transcripts and genes, respectively, in the ovine transcriptome and genome reference. The quality of sequencing provided by the Phred Scores ranged from 32.23 to 36.46, and a mean mapping quality of 42.0 was achieved. These scores were suitable enough for analyzing transcriptional levels and identifying differences in transcripts and genes among the groups.

### 3.2. Characterization of the Ovine Whole Blood Transcriptome

We analyzed the transcriptome of healthy, diseased, and exposed ewes from ordinary flocks to understand how sheep respond to late-stage natural *Cp* infection. Differential gene expression was determined using DESeq2 and Edge R software, which yielded similar results, indicating their validity. Compared with the controls, we identified 253 differentially expressed transcripts (DETs) in the EP, of which 118 were downregulated and 135 were upregulated, and 385 DETs in the EN, of which 226 were downregulated and 159 were upregulated. Data on the differentially expressed transcripts are provided in Appendix A. The identified transcripts were assigned to the 191 annotated DEGs in the EP (102 upregulated and 89 downregulated) and 256 in the EN (106 upregulated and 150 downregulated). The number of differentially regulated transcripts observed in the EN group was more than 13% above the level in the EP group. Regarding the log2 fold change, the CL-positive tested sheep had higher average values (FC = 2.75) and CL-negative sheep had lower (FC = 1.99) values. Under the given experimental conditions, we also analyzed 10 differentially expressed transcripts of five genes between the EN and EP groups.

The numbers of uniquely expressed and overlapping common genes are depicted in a Venn diagram (Figure 1). A total of 86 DEGs were found to be shared between the EN and EP groups, of which 52 were downregulated and 34 were upregulated. Alternatively, we observed 168 unique DEGs in EN animals (70 upregulated vs. 98 downregulated) and 102 unique DEGs in the EP group (68 upregulated vs. 34 downregulated). The volcano plot (Figure 2) shows expression profiling of the transcripts related to ewe infection with *C. pseudotuberculosis*. The Venn diagram and volcano plots illustrate that EN animals had more unique DEGs and DETs, and the share of downregulated genes and transcripts was greater than that in the EP group. In contrast, in the EP group, the number of upregulated genes and transcripts was slightly higher.

### 3.3. Functional Significance of Unique and Common DEGs

The unique and common DEGs observed among the top 10, ranked by their log2 fold change, are presented in Table 2 and Table 3, respectively. In addition, the functional importance of the DEGs and their biological role were screened using the NCBI Gene Resource and UniProt Knowledgebase, considering bovine or human orthologues.

#### 3.3.1. Unique Genes with the Highest DE

The top unique upregulated transcripts in the blood of the diseased ewes included the most abundant killer cell lectin-like receptor subfamily B member 1 (KLRB1, CD161, CLEC5B), which is involved in the native immune response (IR). Other highly expressed genes encode the enzyme kelch-like family member 18 (KLHL18), which regulates proteasomal protein ubiquitination and protein glucosyl galactosyl hydroxylysine glucosidase (PGGHG). This top group further consists of kidney-specific chloride voltage-gated channel (CLCNKA) and immunoglobulin superfamily DCC subclass member 3 (IGDCC3). Vascular cell adhesion protein 1 (VCAM-1) participates in various processes of adaptive immunity, and phospholipid phosphatase related 5 (PLPPR5) regulates neurite biosynthesis.

The top unique downregulated genes in diseased ewes were represented by extracellular matrix (ECM) proteins and organizators, namely, asporin (ASPN), which showed the lowest expression (FC = −6.7), secreted protein acidic and cysteine rich (SPARC), and secreted phosphoprotein 1 (SPP1). Decreased expression was further measured in twist family bHLH transcription factor 1 (TWIST1), which regulates tissue development and fat metabolism, and solute carrier family 7 member 11 (SLC7A11), which enables the transport of small amino acids.

In the exposed group, the top unique genes with elevated expression were more functionally diversified and showed considerably less expression change (FC of 2.41 to 3.45) than in diseased animals. The most abundant transcripts were identified as a ubiquitous transcription factor, zinc finger protein 227 (LOC101109384), a multidomain ankyrin repeat domain 7 protein (ANKRD7) whose function is not yet fully recognized, and an eye component crystallin gamma S (CRYGS). In addition, immune-relevant highly expressed transcripts were identified as ubiquitin ligase tripartite motif containing 45 (TRIM 45), which may act as a transcriptional regulator in the mitogen-activated protein kinase signaling pathway (MAPK) and a pathogen pattern-recognition receptor of T lymphocytes—antigen WC1.1.

Conversely, the top downregulated genes showed higher FC in the negative sheep than in the positive sheep, ranging from 3.8 to 7.44. The unique least expressed genes were large structural proteoglycan 4 (PRG4), an activator of Wnt signaling secreted frizzled-related protein 1 (SFRP1), and a heme protein myeloperoxidase (MPO). Among the low-abundance transcripts, we further observed two ECM components containing collagen: tubulointerstitial nephritis antigen-like 1 (TINAGL1) and collagen type VI alpha 1 chain (COL6A1).

#### 3.3.2. Common Genes with the Highest DE

The common and highly differentially expressed genes are presented in Table 3. The highest expression with the same direction was observed for these immune-relevant DEGs: MHC class I polypeptide-related sequence B (MICB), lymphocyte antigen 6 family member G6C (LY6G6C), related to MHC class III, and hemicentin 1 (HMCN1). On the other hand, the microfibril-associated protein 5 (MFAP5), proteoglycan syndecan 2 (SDC2), and mesenteric estrogen-dependent adipogenesis (MEDAG) transcripts were identified as being the least numerous in the tested blood samples of the CL-affected animals, and had the lowest gene expression values in the study in terms of FC. Only two common genes with the opposite direction of expression between diseased and exposed CL groups were mapped and predicted as MHC class I receptor—leukocyte immunoglobulin-like receptor subfamily A member 6 (LILRA6), involved in adaptive IR, and immuno-associated protein GTPase IMAP family member 7 (GIMAP7).

### 3.4. Enrichment of Gene Ontology (GO) Terms and Pathways in the Investigated Ovine Groups

#### 3.4.1. Cellular Localization of Differentially Expressed Genes

To explain the molecular function of identified DEGs that may be related to the ovine CL, the particular gene clusters connected with different levels of expression within the EP and EN ewes were interpreted using the WebGestalt tool. The results of the cellular component enrichment are depicted in Figure 3. In the positive group, the gene list contains 191 DEG IDs, of which 184 IDs were unambiguously mapped to unique Entrez genes, and 134 gene IDs (70.2%) could be annotated to the enriched cellular components (CC). Similarly, in the negative group, the list consists of 256 gene IDs, of which 246 IDs were mapped, and 178 of them (69.5%) were annotated. In terms of the affected cellular components, the DE genes in diseased ewes were localized mainly in the cell surface (GO:0009986; FDR 0.003) and endocytic vesicle lumen (GO:0071682; FDR 0.003), followed by the ECM (GO:0031012; FDR 0.005), extrinsic component of the plasma membrane (GO:0019897; FDR 0.006), and endoplasmic reticulum (GO:0005783; FDR 0.05). In exposed ewes, we recorded significant differential gene expression in collagen-containing ECM (GO:0062023; FDR 0.0006), ECM (GO:0031012; FDR 0.004), and supramolecular fiber (GO:0099512; FDR 0.02).

#### 3.4.2. Influenced Biological Processes and Role of DEGs in the Host Immune Response

The examined ewes did not show statistically significant enrichment of the classified biological processes with ongoing *Cp* infection. Nevertheless, the top-ranked categories, according to their FDR value, are presented in Figure 4. In the case of the EP group, the GO Slim summary was based on 160 DEGs (85.1% of all DEGs of the EP group) that could be successfully mapped to the genome reference and annotated to the categories. In the EN group, 215 DEGs (85.3%) could be classified. After redundancy reduction, the most enriched in the positive ewes were immune response (GO:0006955; FDR 0.2) and regulation of the immune system process (GO:0002682; FDR 0.3). Among the potentially most influenced processes, we further identified the organonitrogen compound catabolic process (GO:1901565; FDR 0.6), response to cytokine (GO:0034097; FDR 0.6), and ECM organization (GO:0030198; FDR 0.6). In negative ewes, we mainly observed enrichment of the cell cycle (FDR 0.07; GO:0007049), immune response (FDR 0.5), regulation of cell death (GO:0010941; FDR 0.7), and positive regulation of transport (GO:0051050; FDR 0.7), particularly norepinephrine transport (GO:0015874; FDR 0.4).

The ewes’ immune response may include up to 51 immunomodulating DEGs in the EP and EN groups; 20 genes are shared between the groups. Using GO analysis, we further refined their most likely biological function during the interaction with *Cp* in the infected flock. The involvement of immune genes in specific biological processes, ranked according to their enrichment ratio, is detailed in Table 4.

In positive, diseased animals, we observed the influence of biological processes that lead, on the one hand, to activation and positive regulation of the immune cells and molecules and, on the other hand, to the deregulation of some effector components of the sheep defense system. In particular, regulation of antigen receptor-mediated SP and B-cell receptor SP, adaptive IR based on somatic recombination of immunoglobulin receptors, response to other organisms, and some processes of innate IR involved mainly more transcribed DEGs. Humoral IR, complement activation, and homeostasis of calcium ions showed lowered transcriptional activity. The momentary cytokine-mediated inflammatory response measured in the blood transcriptome of the diseased sheep might be lower, but cytokine regulation itself was well balanced. Cytokine signaling accounted for approximately half of all DEGs assigned to the immune response in diseased sheep.

In the exposed sheep, we observed almost the same number of IR genes that were differentially regulated as in the positive animals. Unlike the positive animals, the majority of immune DEGs showed limited transcriptional activity in the corresponding BP. Reduced gene expression was recorded for part of innate IR and regulation of the protein metabolic process. Upregulated DEGs preferentially act in the positive regulation of natural killer cell-mediated cytotoxicity. Furthermore, they positively participate in I-kappa B kinase/NF-kappa B signaling and gamma-delta T-cell activation. Apoptosis and stress response were both positively and negatively regulated in exposed animals. Unlike positive, diseased animals, higher transcription of the genes driving cellular calcium ion homeostasis was observed.

#### 3.4.3. Differential Gene Expression in Biological Pathways Influenced by *C. pseudotuberculosis*

To understand the main potentially affected pathways, the Kyoto Encyclopedia of Genes and Genomes (KEGG), Reactome [29], and WikiPathways [30] databases were applied under the same conditions as the GO analysis. Analysis was subsequently performed on all 447 predicted DE genes across the comparisons among the diseased, exposed, and healthy control sheep (CN × EP and CN × EN). As the table demonstrates, functional enrichment of the pathways following the results obtained in the BP classification of both experimental groups corresponded with the revealed CC. The share of successfully annotated DEGs in the three tested databases was rather low, from 39.9% to 56.4% in the EP group and 41.3 to 52.4% in the EN group. In this respect, Reactome provided the best coverage of the terms through all comparisons. An overview of the main pathways with altered gene expression that met at least the *p* < 0.005 criteria is highlighted in Table 5. We investigated additional pathways that CL affects in the positive, diseased ewes that are often connected with the immune system. The obtained results show significant DE of the “Binding and Uptake of Ligands by Scavenger Receptors” pathway. Highly different transcript abundances were observed for genes maintaining cell and tissue structure and dynamic cooperative signaling in the “ECM organization” and “ECM proteoglycans” pathways. Transcriptional regulation of the “Immunoregulatory interactions between a Lymphoid and nonlymphoid cell” pathway seems to be substantially changed in the EP as well as in the EN group. Other important, possibly highly altered pathways in the EP group include “Cytokine signaling” and “Immune System”.

In the exposed ewes, significant DE was observed for p53 signal transduction events (FDR 0.04) and correlated TP53 regulation of cell death gene pathways. TP53 (tumor suppressor protein 53) controls the transcription of genes involved in the cell cycle and regulates the expression of a number of genes involved in the intrinsic apoptosis pathway triggered by cellular stress. Consistent with previous DE results in the cellular component category, differential regulation of gene expression in the “Cell cycle” and “Syndecan interactions” was identified.

### 3.5. Validation of RNA-Seq Data by RT–qPCR

To confirm the transcriptome data, four target DEGs with the opposite expression (*MFA5*, *MEDAG*, *HMCN1*, and *LY6G6C*) and three reference genes with very suitable geNorm expression stability (*B2M*, *POLR2A*, and *PPIB*) were analyzed by RT–qPCR. The same RNA samples used for RNA-seq were used for qRT–PCR. We evaluated the normalized relative expression levels measured in the downregulated *MEDAG* and *MFAP5* genes and upregulated *HMCN1* and *LY6G6C* genes. The comparative analysis revealed the consistent trend of the results obtained by both methodological approaches and confirmed the validity of DE measured in the blood of experimental and control sheep (Figure 5).

## 4. Discussion

We first describe the potential influence of gene expression on difficult-to-treat caseous lymphadenitis through the analysis of the blood transcriptome in sheep. In contrast to previous animal model studies conducted under controlled experimental conditions in mice, goats, or cell cultures [18,31,32,33,34], we measured gene expression in sheep raised under field conditions. This approach yielded unique insights into the host–pathogen relationship but also has certain limits. Despite the comprehensive data, transcriptome analysis can only offer limited information about metabolic and health status during sampling. In addition, soluble immune molecules often act briefly, locally, at low concentrations and can therefore be challenging to analyze in the blood. It is important to note that whole blood is a complex mixture of various cells and may not necessarily reflect molecular events in organs [35]. Nevertheless, the alteration of gene expression in peripheral blood may be specific to disease progression and used for pathophysiology identification [36]. Understanding how the host and pathogen interact in the initial phases of a disease is crucial to comprehending the infection’s mechanism. However, transcriptomic analysis of the later phases may guide changes occurring in the preclinical and clinical phases of the disease [37].

In this study, we highlight the value of RNA-seq in monitoring genes that could modify the immune response during infection. Our findings show that the blood transcriptome differences between the two experimental groups were less pronounced. Several authors have observed that the full onset of CL can be preceded by an incubation period in which antibodies are not detectable, and some animals may not show them at any phase after the infection [38,39,40]. There is also evidence of a slight difference in the blood transcriptome between infected *Mycobacterium avium* subsp. *paratuberculosis* (MAP) and exposed animals belonging to paratuberculosis-positive herds [37,41]. The phylogenetically related genera *Mycobacterium* and *Corynebacterium* probably share some pathological features due to their intracellular nature [4,7].

A survey of DEGs observed in the experimental groups discussed below is presented in Table 2, Table 3 and Table 4. Compared to the previous transcriptome measure of early exposure to *Cp* in goats, [21] the number of differentially expressed genes was substantially lower in our study. Additionally, the changes in metabolic processes and pathways in the later phase of infection were less pronounced. The host immune mechanisms appeared to be relatively balanced in controlling the adverse effects of infection, especially in diseased ewes. The number of differentially expressed genes in infected animals often decreases with time [42].

A member of the Ig superfamily, VCAM-1 (vascular cell adhesion molecule), was identified as the main differentially expressed unique gene between diseased and exposed ewes. It is a cell surface glycoprotein expressed by the cytokine-activated endothelium that helps regulate inflammation-associated vascular adhesion and the transendothelial migration of macrophages and T cells. The expression of VCAM-1 is activated by proinflammatory cytokines, and, together with integrins, it plays a central role in leukocyte recruitment during inflammation [43]. Its high concentration in the blood of diseased animals may indicate active regulation of the inflammatory response to ongoing *Cp* infection.

The most abundant and unique DE gene investigated in the EP group compared to healthy controls was KLRB1 (killer cell lectin-like receptor subfamily B member 1, CD161), which encodes a receptor expressed on natural killer (NK) cells and peripheral blood memory T cells. It defines a functionally distinct subset of proinflammatory NK cells and marks cells that have retained the ability to respond to innate cytokines during their differentiation. It is interesting to note that upregulation of KLRB1 was also found in the whole blood of MAP-positive cattle [44]. The most downregulated gene in diseased animals detected using RNA-seq was a common gene for microfibril-associated protein 5, MFAP5. In cattle, it is involved in definitive hemopoiesis, and its deficiency in mice is connected with decreased levels of neutrophils in circulation [45].

The highly abundant transcript among exposed animals was hemicentin, HMCN1. This gene encodes a conserved large ECM protein, a member of the Ig superfamily [46]. It is involved in cell adhesion, cytoskeleton organization, and response to bacteria. On the other hand, mesenteric estrogen-dependent adipogenesis, MEDAG, was examined as the most downregulated transcript. This gene is believed to promote adipocyte differentiation, lipid accumulation, and glucose uptake in mature adipocytes, but its role in immunity is yet to be determined [47]. It is also noteworthy that the transcriptional activity of the PRG4 gene was deficient in the EN group. It has been proposed that PRG4 is regulated by the inflammatory response, typically in a negative fashion, and once the inflammation has subsided, PRG4 returns to homeostatic levels to maintain tissue health [48].

The infection primarily interfered with the expression of many sensing surface receptors. Various other differentially expressed C-lectin receptors of NK cells and macrophages were identified, such as CLECL1, CLEC2D, and CLEC4E (Mincle). Recently, it was discovered that Mincle (macrophage-inducible Ca2+-dependent lectin) acted as a receptor for corynomycolic acids of the corynebacterial cell wall. It probably cooperates with other C-lectin receptors in *Cp* recognition, activation of immune cells, and modulation of the proinflammatory response to prevent possible tissue damage [49,50,51]. In addition to the C-lectin receptors, RNA-seq analysis identified several G protein-coupled receptors and six different transcripts of LIR receptors. Ig-like receptors LIR, LILRA5, and LILRA6 of NK cells showed DE in both experimental groups, which was more pronounced in exposed ewes. They drive leukocyte activity and play an essential role in regulating the immune response. The significance of LIR receptors in the innate immune response has been recently broadened to include the ability to develop antigen-specific immune memory [52].

The category of peptide-presenting antigens to NK receptors was almost highly upregulated in affected ewes. It consists of two classical BOLA class I histocompatibility antigens, OVAR and OLA-I, and MHC-III lymphocyte antigen 6 family member G6C, LY6G6C, and nonclassical MHC proteins and cooperating molecules, MHC class I polypeptide-related sequence B, MICB, UL16-binding protein 2-like, ULBP2, and butyrophilin subfamily 2, member A2, BTN2A2. Many studies have confirmed the pivotal role of MHC molecules in antigen presentation required to elicit an immune response against invading pathogens. The expression of MHC genes is altered during mycobacterial infection in cattle [42,53,54]. In transcriptome analysis of the *Cp*-infected spleen of dairy goats, the MHC class I protein complex was identified as one of the most enriched terms [21]. The genes for CD1A and CD1E molecules structurally related to MHC, which present lipids of microbial origin to T-cell receptors, were differentially expressed only in the transcriptome of the diseased ewes. The CD1 system is considered a versatile player in the immune response, sitting at the crossroads of innate and adaptive immunity, which may be involved in numerous infectious and inflammatory responses [55].

Our study found that both experimental groups had a decreased abundance of GZMA, GZMB, and GZMH transcripts in their blood. These transcripts are part of the granzyme serine protease family, which is specific to cytolytic T lymphocytes and NK cells. Proteases directly contribute to immune defense by inducing cell cytolysis and pathogen clearance in innate immunity [56]. GZMA is a potential biomarker of human *Mycobacterium tuberculosis* (MT) infection and disease, as its serum levels are significantly lower in patients with active MT disease [57]. Additionally, GZMA may be captured and internalized within mycobacteria-infected monocytes in influenced organs to inhibit bacterial growth and prevent infection [58]. Based on our findings, we hypothesize that granzyme levels may fluctuate in the peripheral blood of CL-affected animals and that its lower transcription can be attributed to a particular disease stage. The family of GIMAP enzymes (specifically GIMAP1, GIMAP7, and GIMAP8) was expressed in either the same or different directions in both diseased and exposed sheep. GTPases of immunity-associated proteins are regulators of lymphocyte survival and homeostasis and are linked to inflammatory and autoimmune diseases [59,60]. Recent studies suggest that GIMAP proteins may interact with each other and be involved in the movement of the cellular cargo along the cytoskeletal network [61].

Several authors have emphasized the significance of the cellular immune response and proinflammatory cytokines in limiting bacterial growth and dissemination inside the host in the initial phase of *Cp* infection [18,62]. Through transcriptome analysis of ovine blood samples, we confirmed the involvement of T lymphocyte-mediated immunity. Gamma-delta T-cell activation predominates in exposed ewes, along with the expression of their specific WC1.1 antigen. γδ T lymphocytes are thought to play crucial roles in immunosurveillance and host defense, particularly against mycobacteria [63]. Glycoprotein WC1.1 is highly represented in ovine peripheral blood and acts as both a γδ T-cell activating coreceptor and pattern-recognition receptor [64]. In goat cell culture, only WC1(+) γδ T cells were able to produce proinflammatory cytokines [65]. We also observed increased regulation of innate proinflammatory NF-kappa B signaling in the EN group. In contrast, a detailed analysis of the goat transcriptome revealed downregulation of the NF-κB pathway in *Cp*-infected individuals [21]. However, bovine MAP-infected macrophages produce larger amounts of NF-κB during the initial phase of infection [66]. Many DEGs (approximately 30) in exposed animals are involved in programmed cell death and response to cellular stress as part of normal physiological processes and native immunity. The regulation of their expression tends to be balanced, probably to maintain homeostasis during infection and to prevent excessive cell damage. It has been reported that apoptosis is an essential defense mechanism for host resistance to pathogen invasion. However, *Cp* infection likely contributes to elevated programmed cell death by apoptosis [21]. Accordingly, the death of host cells colonized by corynebacteria has been observed to occur more frequently during the early stage of the pathogenic process [67].

In our study of positive sheep with caseous lymphadenitis, we observed an increase in the regulation of the adaptive immune response through signal transduction and cytokine signaling. The pathogenesis of ovine caseous lymphadenitis was shown to be associated with the production of cytokines at the pyogranuloma level. However, the local cytokine patterns associated with different courses of infection and the time following infection were not distinguished [2,34]. We found both positively and negatively regulated host responses to cytokines but partly lowered cytokine and chemokine production and its regulation. During this phase of the disease, lower expression of chemokine CXCL10, chemokine receptor CXCR6, complement protein C4BPA, complement receptor C3AR1, and anti-inflammatory interleukin IL13 suggests a limited inflammatory response, complement activation, chemotactic activity, and lymphocyte migration toward the inflammation site. Likewise, MAP has the ability to downregulate the host complement, hence increasing the chance of survival of bacteria when outside the host cells [53]. Recently, C-X-C motif chemokine ligand 10 has been proposed as a possible biomarker of bovine tuberculosis with the potential to differentiate between latent and active disease. Due to the granulomatous nature of mycobacteria and also related corynebacteria, chemokine recruitment of leukocytes may be a host response to contain invading bacteria, and the restriction of this process by bacteria can subvert the host immune response and establish a latent infection [54]. We suppose that the mechanism behind this phenomenon may be the ability of phospholipase D to reduce leukocyte migration toward sheep serum [68].

Although the transcriptional regulation of the humoral IR appeared to decrease, B-cell activation and B-cell receptor signaling orchestrated by the highly expressed receptor TNFRSF13C (TNF receptor superfamily member 13C) were upregulated in diseased ewes. This receptor is considered the principal receptor required for B-cell-activating factor (BAFF)-mediated mature B-cell survival [69]. The initial immune response to *Cp* has been shown to include a strong humoral component, but little is known about the kinetics of protective antibodies [62,70]. After experimental infection in goats, the humoral IR showed individual intensity variations and regularly declined over time [20]. The positive regulation of the adaptive immune response based on somatic recombination of immune receptors built from Ig superfamily domains indicates the potential of the host to develop both clonal-specific antibodies against *Cp* and protective immunological memory during disease progression. We suggest that this can contribute to more effectively eliminating bacteria that can escape from granulomas and spread through the lymphatic system.

We identified significant enrichment of several pathways that may be altered in relation to ongoing infection. In diseased animals, the only statistically significant pathway of binding and uptake of ligands by scavenger receptors was recognized. Scavenger receptors bind free extracellular ligands as the initial step in their clearance from the body. They represent an important part of the innate immune defense by acting as bacterial pattern-recognition receptors. The receptors can modulate the inflammatory response as coreceptors of Toll-like receptors, and some intracellular pathogens utilize them to enter host cells [71]. Pathways of ECM proteoglycans and ECM organization containing significant amounts of DEGs were also highly differentially regulated in our research, especially in a negative way. ECM, as a major component of the cellular microenvironment, influences cell adhesion and migration, and its composition is highly heterogeneous and dynamic, being constantly remodeled [72]. Therefore, we suppose the current infection can degrade the essential immune and adhesive functions of ECM and its large protein structures. In accordance with our results, transcriptional profiling of Holstein cows yielded similar findings on ECM modifications in response to natural MAP infection [73].

We also observed substantial expression differences in the immunoregulatory interactions between lymphoid and nonlymphoid cell pathways, which play a critical role in modifying the response of lymphoid cells to pathogenic organisms. Lymphoid cells are able to regulate their location and movement in accordance with their state of activation and home in tissues expressing the appropriate complementary ligands [74]. The pathway involves mainly identified NK immune receptors (LILRs and KIRs) and adhesion molecules (i.e., VCAM-1) necessary for antigen presentation by MHC and cooperating molecules (i.e., MICB) and maintaining immunological synapses [75]. It is transcriptionally differentially regulated in both experimental groups facing infection compared to healthy animals.

The p53 pathway was found to be the only one significantly affected in exposed ewes. This finding aligns with the significant transcriptional regulation of genes related to cell death and cell cycle. The p53 transcription factor, known as the tumor suppressor protein, is the key gatekeeper in the cellular response to stress signals. It can help exposed animals respond adequately to stress and prevent DNA damage that is potentially caused by infectious pressure. It also allows cells to recover from damage and survive, preventing premature cell death.

In conclusion, we want to draw attention to some of the limitations or shortcomings of this study that need to be considered. First, to improve the accuracy of the results, it may be necessary to work with a larger sample size during CL infection and exposure. Additionally, due to the incomplete functional annotation of sheep genes, it was necessary to rely on bovine or human homologs, which prevented the full interpretation of some discovered genes. Last, the differences in gene expression levels we observed between the investigated ewe groups could be affected by other factors, such as the animal’s physiological status, management or environmental differences, way and frequency of deworming, and resistance to deworming medication.

## 5. Conclusions

We performed the first comprehensive analysis of the peripheral blood transcriptome in sheep naturally infected with *Corynebacterium pseudotuberculosis* from an ordinary flock. Bacterial infection interfered with host gene expression, which was reflected preferentially in the transcriptional regulation of genes that modulate the immune response. Two investigated groups (exposed—negative group and diseased—positive group) of adult East Frisian ewes displayed similar transcriptome patterns during longer contact with pathogens but differences in gene expression compared to healthy animals. Hundreds of sheep DEGs associated with the pathogenesis of caseous lymphadenitis, major cellular components, biological processes, and pathways potentially impacted by the disease were identified. Although it is difficult to unambiguously ascribe these differential gene products as contributors to the pathogenesis of caseous lymphadenitis, this study has identified genes not previously associated with CL exposure in sheep. The most differentially expressed genes were involved in lymphocyte function, MHC antigen presentation, and extracellular matrix organization. Several major biological processes appear to be triggered in response to CL, including increased activation of the adaptive immune response, altered cytokine production and signaling, and suppression of the inflammatory response. In addition, exposed ewes tended to have notably influenced innate immune processes and programmed cell death. This study suggests that the pathway of antigen-binding scavenger receptors might play a substantial role in the protective immune response against *Cp*. It is not precisely known whether the pathogen, the host, or both drive the changes in gene expression described in *Cp*-infected sheep. It would be interesting to hypothesize that the differences in expression may provide information about biomarkers that distinguish the preclinical stage of CL; however, further research will be necessary to explore the roles of particular genes in the disease course and the involved mechanisms.

## Figures and Tables

**Figure 1 animals-13-02144-f001:**
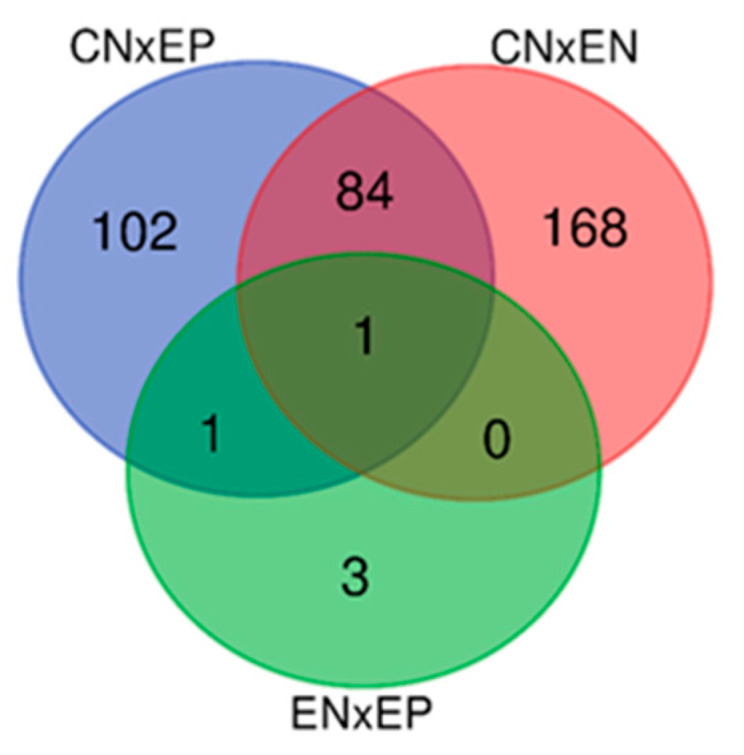
Venn diagram comparing the number and overlapping relationships of DE genes between the diseased (EP), exposed (EN) and control (CN) groups. The overlapping regions indicate shared DE genes of two or three comparable groups.

**Figure 2 animals-13-02144-f002:**
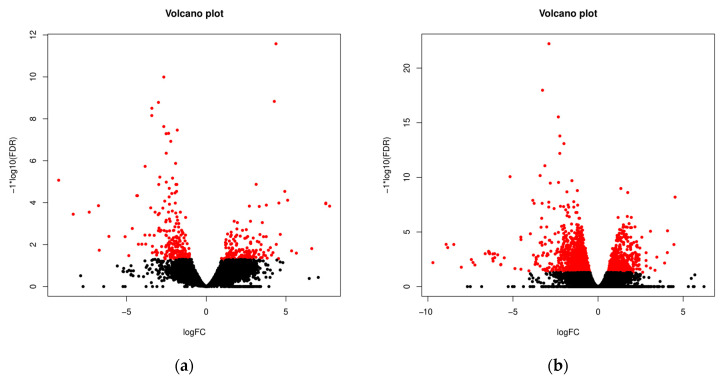
Volcano plots. The red color indicates transcripts detected as differentially expressed between the diseased (EP) and control groups (CN) (**a**) and between the exposed (EN) and control groups (CN) (**b**) at FDR < 0.05. Positive *x*-values represent upregulation, and negative *x*-values represent downregulation.

**Figure 3 animals-13-02144-f003:**
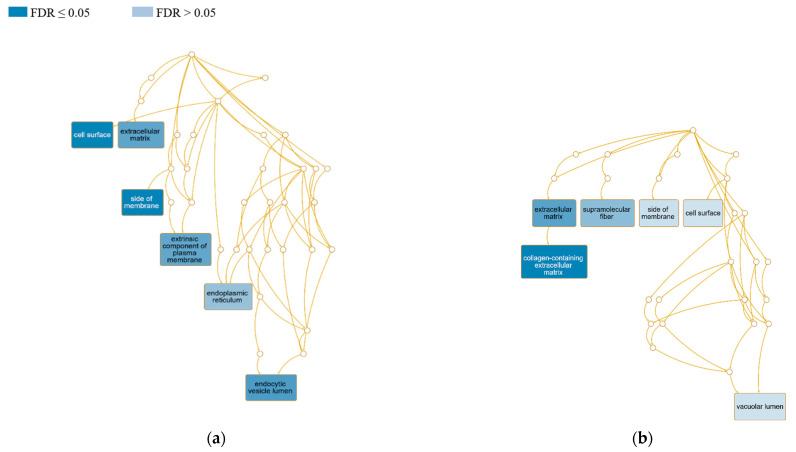
Scheme of the relationships among the CL-influenced cellular components in positive and negative sheep compared to healthy controls by means of a directed acyclic graph (DAG) generated using the Webgestalt tool. The graph was constructed based on the current hierarchy of GO terms www.geneontology.com (accessed on 30 April 2023), where each GO term is a node, and the relationships between the terms are edges between the nodes. Redundancy reduction in GO terms was applied through the weighted set cover (WSC) approach in compact mode. (**a**) CL-positive and (**b**) CL-negative ewes.

**Figure 4 animals-13-02144-f004:**
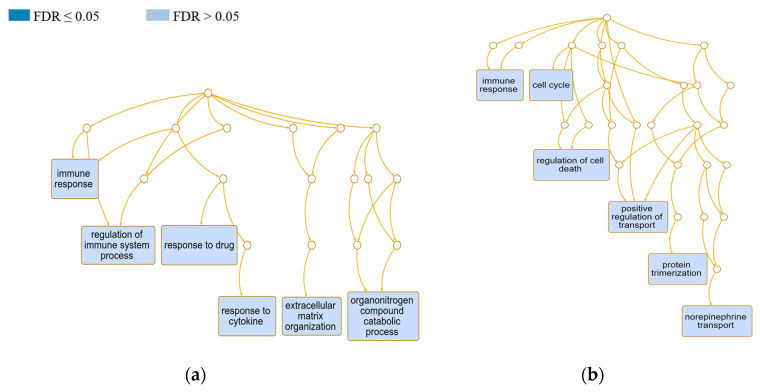
Scheme of the relationships among the CL-influenced biological processes in positive and negative sheep compared to healthy controls by means of a DAG generated using the Webgestalt tool. The graph was constructed based on the current functional hierarchy of GO terms www.geneontology.com (accessed on 30 April 2023), where each GO term is a node, and the relationships between the terms are edges between the nodes. The graph illustrates the top-ranked categories with the lowest FDR values. Redundancy reduction in GO terms was applied through the WSC approach in compact mode. (**a**) CL-positive and (**b**) CL-negative ewes.

**Figure 5 animals-13-02144-f005:**
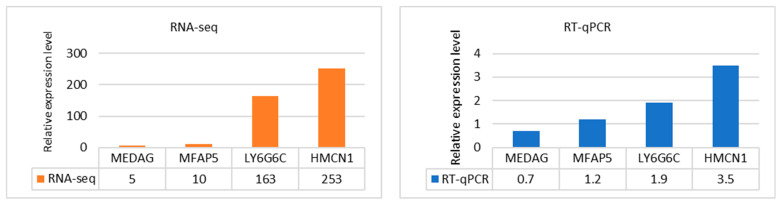
RT–qPCR validation of the RNA-seq expression data. The bar diagrams show comparable trends in the relative expression levels of the four tested DEGs.

**Table 1 animals-13-02144-t001:** Sequencing and mapping statistics of the sequencing reads—mean values.

Group	Trimmed Reads	Mapped Reads	Mapped Reads %	Total Paired Reads	Uniquely MR	Uniquely MR %	Transcripts	Genes
EP	34,454,888	26,542,080	77.02	25,825,876	14,965,299	56.34	27,430	12,230
EN	30,881,461	24,175,800	78.29	23,691,710	13,300,932	55.02	26,730	12,080
CN	30,495,644	19,935,851	63.64	19,501,543	11,341,457	57.03	26,753	11,932

Abbreviations: CN, control group of healthy ewes; EN, negative, exposed ewes; EP, positive, diseased ewes; MR, mapped reads.

**Table 2 animals-13-02144-t002:** The top 10 unique DE genes for *Cp*-affected sheep as ranked by fold change.

Gene Name	Gene Symbol	FC	*p* _adj_
CNxEP			
Killer cell lectin-like receptor subfamily B member 1	*KLRB1*	7.74	1.46 × 10^−4^
Kelch-like family memeber 18	*KLHL18*	6.62	1.53 × 10^−2^
Chloride voltage-gated channel Ka	*CLCNKA*	5.35	1.98 × 10^−2^
Immunoglobulin superfamily DCC subclass member 3	*IGDCC3*	5.11	7.60 × 10^−5^
Protein glucosyl galactosyl hydroxylysine glucosidase	*PGGHG*	4.64	3.25 × 10^−3^
Asporin	*ASPN*	−6.71	1.84 × 10^−2^
Secreted protein acidic and cysteine rich	*SPARC*	−4.36	4.60 × 10^−5^
Secreted phosphoprotein 1	*SPP1*	−4.32	4.62 × 10^−5^
Solute carrier family 7 member 11	*SLC7A11*	−3.21	2.24 × 10^−2^
Twist family bHLH transcription factor 1	*TWIST1*	−3.14	1.84 × 10^−2^
CNxEN			
Zinc finger protein 227	*LOC101109384*	3.45	1.99 × 10^−3^
Ankyrin repeat domain 7	*ANKRD7*	2.82	7.95 × 10^−4^
Crystallin gamma S	*CRYGS*	2.62	4.84 × 10^−2^
Antigen WC1.1	*WC1.1*	2.42	2.12 × 10^−3^
Tripartite motif-containing 45	*TRIM45*	2.42	2.50 × 10^−2^
Proteoglycan 4	*PRG4*	−7.44	3.33 × 10^−3^
Secreted frizzled-related protein 1	*SFRP1*	−7.35	6.14 × 10^−3^
Tubulointerstitial nephritis antigen-like 1	*TINAGL1*	−4.53	4.98 × 10^−5^
Collagen type VI alpha 1 chain	*COL6A1*	−4.06	3.62 × 10^−2^
Myeloperoxidase	*MPO*	−3.80	1.30 × 10^−3^
ENxEP			
Vascular cell adhesion protein 1	*VCAM1*	7.50	2.68 × 10^−2^
Phospholipid phosphatase related 5	*PLPPR5*	6.12	2.57 × 10^−2^

Abbreviations: DE, differentially expressed; CN, control group of healthy ewes; EN, negative, exposed ewes; EP, positive, diseased ewes; FC, log2 fold change; *p*_adj_, adjusted *p*-value.

**Table 3 animals-13-02144-t003:** The top 10 common DE genes for *Cp*-affected sheep as ranked by fold change.

Group		EP		EN	
Gene Name	Gene Symbol	FC	*p* _adj_	FC	*p* _adj_
Same direction of expression					
Hemicentin 1	*HMCN1*	4.93	2.89 × 10^−5^	4.07	7.72 × 10^−4^
MHC class I polypeptide-related sequence B	*MICB*	4.37	2.63 × 10^−12^	4.52	6.42 × 10^−9^
Lymphocyte antigen 6 family member G6C	*LY6G6C*	3.77	1.30 × 10^−4^	4.08	7.84 × 10^−6^
Tyrosine protein phosphatase nonreceptor-type substrate 1	*PTPN1*	3.59	4.17 × 10^−3^	3.91	6.92 × 10^−3^
Microfibril-associated protein 5	*MFAP5*	−9.26	8.45 × 10^−6^	−6.63	9.73 × 10^−4^
Syndecan 2	*SDC2*	−8.35	3.52 × 10^−4^	−5.72	1.00 × 10^−2^
Mesenteric estrogen-dependent adipogenesis	*MEDAG*	−7.34	2.83 × 10^−4^	−8.93	1.36 × 10^−4^
Decorin	*DCN*	−6.76	1.39 × 10^−4^	−5.51	2.31 × 10^−3^
Opposite direction of expression					
Leukocyte immunoglobulin-like receptor subfamily A member 6	*LILRA6*	−1.72	9.82 × 10^−4^	3.08	8.59 × 10^−6^
GTPase IMAP family member 7	*GIMAP7*	−1.50	2.43 × 10^−3^	2.57	3.02 × 10^−5^

Abbreviations: EN, negative, exposed ewes; EP, positive, diseased ewes; FC, log2 fold change; *p*_adj_, adjusted *p*-value.

**Table 4 animals-13-02144-t004:** Involvement of the immune-modulating DEGs in biological processes according to the enrichment ratio.

DEGs Ordered According to Their FC	Biological Processes	
EP	Upregulated	Downregulated
*KLRB1*; *DCN*; *IGDCC3*; *HMCN1*; *PRG3*	Antigen receptor-mediated SP	Humoral IR
*ADAMTS13*; *MICB*; *SPARC*; *SPP1*; *IL13*	B-cell receptor signaling pathway	Cytokine production
*LY6G6C*; *PTPN1*; *OLA-I*; *COL1A1*; *BTN2A2*	Response to other organisms	Calcium ion homeostasis
*GZMH*; *LAMB1*; *TNIP2*; *GATA6*; *SPIB*	Adaptive IR somatic recomb. of IRCs built from Ig domains	Inflammatory response
*GIMAP8*; *NFKB2*; *C4BPA*; *GZMA*; *IBSP*	Response to cytokine	Complement activation
*TNFRSF13C*; *SH2B2*; *PAX5*; *BLK*; *MT2A*	Cell adhesion	
*CXCR6*; *JCHAIN*; *IFITM1*; *CXCL10*; *UBD*	Lymphocyte activation	
*C3AR1*; *FCRL3*; *CDA*; *GHR*; *UBA7*; *CLECL1*	Innate IR	
*ADGRE3*; *CD1A*; *HSP90B1*; *LILRA6*; *OVAR*		
*BPI*; *ULBP1*; *CD1E*; *GIMAP7*		
**EN**		
*PRG4*; *DCN*; *MICB*; *LY6G6C*; *HMCN1*; *PTPN1*	Positive regulation of NK cell-mediated cytotoxicity	Innate IR
*MPO*; *COL1A1*; *IL13*; *GZMB*; *GIMAP8*	Gamma-delta T-cell activation	Regulation of response to stress
*C4BPA*; *SERPINB4*; *BMPR1A*; *LAMB1*	I-kappaB kinase/NF-kappaB signaling	Regulation of protein metabolic process
*GIMAP1*; *LILRA6*; *DQB*; *CLEC4E*; *IL1RN*	Response to stress	Apoptotic process
*GIMAP7*; *TRIM45*; *DNM1*; *GZMH*; *HMGB2*	Apoptotic process	Locomotion
*PRG3*; *SOX13*; *C3AR1*; *GZMA*; *ULBP2*	Calcium ion homeostasis	
*WC1.1*; *CCL14*; *DNM3*; *NLRP2*; *TNFSF10*		
*GCA*; *KIR3DP1*; *LILRA5*; *BLK*; *UBD*; *PILRB*		
*SERPINB9*; *CD226*; *FCER1A*; *ADGRE1*; *OLA-I*		
*CLEC2D*; *TNFRSF25*; *GHSR*; *PSPC1*; *BPI*		

Abbreviations: DEGs, differentially expressed genes; EN, negative, exposed ewes; EP, positive, diseased ewes; FC, log2 fold change; Ig, immunoglobulin; NK, natural killer; IR, immune response; IRCs, immune receptors.

**Table 5 animals-13-02144-t005:** Enrichment of *Cp*-influenced pathways in the investigated groups of affected ewes.

Gene Set	Description	Size	Overlap	ER	*p*-Value	FDR
EP						
R-HSA-2173782	Binding and uptake of ligands by scavenger receptors	42	6	14.2	3.48 × 10^−6^	0.01
R-HSA-3000178	ECM proteoglycans	76	6	7.9	1.10 × 10^−4^	0.06
R-HSA-1474244	Extracellular matrix organization	301	11	3.6	2.11 × 10^−4^	0.06
R-HSA-1480926	O_2_/CO_2_ exchange in erythrocytes	13	3	21.0	2.62 × 10^−4^	0.06
R-HSA-198933	Immunoregulatory interactions between a lymphoid and a nonlymphoid cell	132	8	5.3	3.56 × 10^−4^	0.08
R-HSA-1280215	Cytokine signaling in the immune system	688	16	2.3	1.33 × 10^−3^	0.25
R-HSA-168256	Immune system	1997	32	1.60	3.35 × 10^−3^	0.40
**EN**						
hsa04115	p53 signal pathway	72	7	6.2	1.23 × 10^−4^	0.04
R-HSA-198933	Immunoregulatory interactions between al and a nonlymphoid cell	132	7	3.7	2.73 × 10^−3^	0.11
R-HSA-1793185	Chondroitin sulfate/dermatan sulfate metabolism	50	4	9.3	8.80 × 10^−4^	0.19
R-HSA-1480926	O_2_/CO_2_ exchange in erythrocytes	13	3	18.5	4.99 × 10^−4^	0.29
hsa04110	Cell cycle	124	7	3.6	3.22 × 10^−3^	0.34
R-HSA-5633008	TP53 Regulates the transcription of cell death genes	44	4	7.3	2.15 × 10^−3^	0.41
R-HSA-3000170	Syndecan interactions	27	3	8.9	4.49 × 10^−3^	0.65

Abbreviations: EN, negative, exposed ewes; EP, positive, diseased ewes; ER, enrichment ratio.

## Data Availability

The data presented in this study are openly available in the National Center for Biotechnology Information Sequence Read Archive, reference number BioProject ID PRJNA984244.

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
