# Peer review of "Comparative Characterization of Immune Response in Sheep with Caseous Lymphadenitis through Analysis of the Whole Blood Transcriptome"

_animals, 2023, doi:10.3390/ani13132144_

Round 1
Reviewer 1 Report
The manuscript “Comparative characterization of immune response in sheep with caseous lymphadenitis through analysis of the whole blood transcriptome” presents interesting data about caseous lymphadenitis in sheep using RNA-seq.
The following comments about methodology should be addressed by the authors.
In the Abstract, regarding the information “The analysis identified 191 annotated differentially expressed genes (DEGs) in the EP group (102 upregulated and 89 downregulated) and 256 DEGs in the EN group (106 upregulated and 150 downregulated)”, was in the comparison between EPxCN and ENxCN? Or considering all comparisons (ENxEP, ENxCN, and EPxCN)?
It was difficult to understand when the comparisons were made among differentially expressed genes (DEG) and when they were among transcripts (DET)? And also when is referring to EP compared to EN, or EP and EN compared to CN.
The statement “Infection influenced numerous immunoregulatory interactions between lymphoid and nonlymphoid cells in both EP and EN ewes” is confusing, because EN are not infected, but exposed. A clear definition of groups and comparisons is necessary. For example, when saying that “We analyzed the transcriptome of healthy and diseased ewes”. EN was treated as diseased or healthy?
At some point of the manuscript, the CN was left behind and it was assumed that only two groups were used (“Our findings show that the blood transcriptome differences between the two experimental groups were less pronounced” and “Two investigated groups (exposed - negative group and diseased - positive group)”).
In the exposed negative (EN), is there any control of exposition? If they are serologically negative, how to be sure that they were really exposed? Or was it was assumed because they are raised in an infected flock?
Comment the implications of using such a small sample size (3 samples per group) in the study. Are the results suitable?
Justify the use of the version 1 of the sheep transcriptome reference (GCF_002742125.1_Oar_rambouillet_v1.0_rna), considering that version 2 is available.
Did the authors check for overexpressed genes before differential expression analysis?
After DE analysis, were gene counts checked (visualization of gene counts in a box plot to assure the absence of outliers and to compare groups)? Was a minimum gene count considered?
Was RT-qPCR used for validation of only the 3 samples per group or additional samples were included? If only the nine samples were used, it consisted in a technical validation. A biological validation, using additional samples, would be more relevant.
Reviewer 2 Report
The authors present a very interesting manuscript through analysis of the whole blood transcriptome, to understand how ewes immune system responds to Caseous lymphadenitis. It is a very well designed and written manuscript with relevant results.
I just have a minor suggestion regarding the presentation of the table 3, it should be located after the description of the results in the text . In addition I would suggest a In addition I would suggest a reduction of the discussion section.
Reviewer 3 Report
General points:
- The study has used whole blood transcriptomics to study a disease caused by an intracellular organsim with spefici accumulation site at lymph node? Was there not possible to include lymph node tissue in the study? the circulating WBCs and residents lymphocytes might have a very different gene expression profile which could contribute to further understanding of the disease.
- Animal in the study were regularly dewormed. The choice of antiparasitic will no doubt have an effect on the blood transcriptome of these animal (parasitic load variance in each sheep or group). This confounding factor could introduce potential bias in the DEG analysis carried out between CN, EP and EN groups. Authors should include the anti-parasitic medication dosage and generic name along with mentioning this fact in the discussion of the paper.
- Not including the parameters used for trimming, mapping, read subsetting etc. is not really an acceptable and FAIR compliance of carrying out Bioinformatic analysis these days. This point requires serious work up throughout this manuscript.
- The RNA-Seq raw data must be made available to public and should be made a condition for acceptance of this publication. Specially that this research is tax-payer's funded and no sensitive breed where used. There is no excuse not to upload the raw data and the used scripts to a free public repository (FigShare, ENA archive, NCBI SRA etc.)
Specific points:
-L84 Please include the reference for this stats.
-L99 Please include previous studies or manufacturer's product details for this ELISA kit.
-L121 Please include the details of 32million/sample metric. If there was any multiplex lane runs, where there any duplicates for each library or they were only sequenced once all to gether in 1 lane. Please add more details.
-L124 Please inlcude the parameters and flags used in trimgalore alongside the version either in the text or as supplementary material.
-L133 Please clarify if the count matrix was at gene level or transcript. Given the rest of the manuscript it seems to be transcript level expression.
-L162 Please include the primer sequences (forward and reverse) for each gene in the supplementary Table1. Also include the genome assembly used for both the RNA-Seq analysis and qPCR probes.
-L189 Please indicate actual MAPQ phred score range or average instead of the word "Excellent"
-Fig3and4 Please explain the relationship between circles and lines as to what they represent. Its currently very little useful information for the reader in the caption. Please expand the caption and explain why there are GO terms with FDR > 0.05 is even included in the figure! Inclusion of terms that are not significantly enriched (by the limit set by the authors i.e. FDR < 0.05) doesn't qualify as results and discussion points.
NA
Round 2
Reviewer 3 Report
Thanks for your prompt revision and taking onboard crucial suggestions. Just one last minor point I believe needs changing:
L129 - The libraries were pooled no "polled"
Please attempt to shorten your sentences (ease of read) and avoid redundant usage of adverbs e.g. automatically processed --> processed, through out the manuscript. An editorial proof-reading prior to final publication is recommended.